# LEARNING TO PROGRESSIVELY PLAN

## ABSTRACT

For problem solving, making reactive decisions based on problem description is fast but inaccurate, while search-based planning using heuristics gives better solutions but could be exponentially slow. In this paper, we propose a new approach that improves an existing solution by iteratively picking and rewriting its local components until convergence. The rewriting policy employs a neural network trained with reinforcement learning. We evaluate our approach in two domains: job scheduling and expression simplification. Compared to common effective heuristics, baseline deep models and search algorithms, our approach efficiently gives solutions with higher quality.

## 1 INTRODUCTION

In recent years, deep reinforcement learning has achieved strong success in scenarios that are well-defined and can be precisely and efficiently simulated (e.g., games like Atari (Mnih et al., 2013)). One typical scenario is model-free control: given the current state (e.g., a ball is falling), a decision is made *reactively* to yield maximal rewards (e.g., move the pedal to bounce the ball back).

In more challenging scenarios, a one-off reactive decision is usually insufficient. It is necessary to *plan* before a strong decision. As illustrated by many works, especially competitive games like Chess and Go (Silver et al., 2017), most of the times, search-based policy is stronger than reactive ones. However, the time consumption can also be exponential, if planning is done from scratch. To address this issue, previous works use neural networks to predict (and execute) an entire plan from scratch, given a complete specification of the problem description (Vinyals et al., 2015; Mao et al., 2016; Graves et al., 2014). While this avoids search, a direct prediction could be very difficult and even impossible in complicated scenarios.

An alternative approach is to start from an existing plan and iteratively improve. When both state and action spaces are continuous, many trajectory optimization techniques have been proposed in the context of robotics and control (MAYNE, 1973; Tassa et al., 2012; Levine & Abbeel, 2014), which use local gradient information to gradually improve the existing plan. On the other hand, with discrete state and action spaces, such approaches become inapplicable due to the indifferentiablity.

In this paper, we propose a novel approach of progressive planning in the discrete space, by iteratively picking a local part of an existing plan and rewriting it. The rewriting decision is made by a neural network, trained end-to-end with reinforcement learning. During the training, the neural network learns to find common local patterns in the current solution and replace them with a better choice, until there is no more improvement.

We apply our approach to two different domains: job scheduling and expression simplification. We show that our rewriting approach is better than strong heuristics of both domains using multiple metrics. For job scheduling, under a controlled setting, we also demonstrate that our model outperforms DeepRM (Mao et al., 2016), a neural network predicting a holistic scheduling plan, by large margins especially when more heterogeneous resources lead to more complicated scheduling.

## 2 RELATED WORK

Trajectory optimization problems in the continuous space have been widely studied in the domain of robotics and control, and many effective techniques have been proposed (MAYNE, 1973; Bradtke et al., 1994; Vrabie et al., 2009; Tassa et al., 2012; Levine & Koltun, 2013; Levine & Abbeel, 2014). Typically, local gradient information is leveraged to gradually optimize the existing trajectory. However, these techniques are not directly applicable to domains with discrete state and action spaces. In this work, we study approaches of iterative refinement for discrete optimization problems.

Job scheduling and resource management problems are ubiquitous and fundamental in computer systems. Various work have studied these problems from both theoretical and empirical sides (Błażewicz et al., 1996; Grandl et al., 2015; Armbrust et al., 2010; Scully et al., 2017; Terekhov et al., 2014; Mao et al., 2016; Chen et al., 2017). In particular, recent work use deep reinforcement learning for job scheduling with a simplified setup (Mao et al., 2016; Chen et al., 2017). Existing work focus on proposing algorithms to construct a schedule from scratch. However, with more complex configurations, it becomes challenging to make an effective arrangement in this way, as indicated in our evaluation. Thus, we make an initial step towards tackling the job scheduling problem through rewriting, and we consider extending our approach to real-world settings as future work.

A recent line of work studies using deep neural networks to discover equivalent expressions (Cai et al., 2018; Allamanis et al., 2017; Zaremba et al., 2014). In particular, (Cai et al., 2018) trains a deep neural network to rewrite algebraic expressions. However, they use supervised learning to train the model, which requires a collection of ground truth rewriting paths, and lacks the capability to find novel rewriting routines. On the contrary, we use reinforcement learning to train our rewriter for expression simplification, which mitigates these limitations.

Our rewriting formulation is closely related to other applications such as code optimization (Schkufza et al., 2013; Chen et al., 2018), theorem proving (Huang et al., 2018; Lederman et al., 2018), and text simplification (Cohn & Lapata, 2009; Paetzold & Specia, 2013; Feblowitz & Kauchak, 2013). In addition, our rewriting approach could also be extended to classical combinatorial optimization problems (Khalil et al., 2017; Bello et al., 2016; Vinyals et al., 2015; Karp, 1972), e.g., Traveling Salesman Problem (Applegate et al., 2006) and Vertex Cover Problem (Bar-Yehuda & Even, 1981).

## 3 PROBLEM SETUP

In this work, we propose to formulate optimization as a rewriting problem, and find the optimal solution through an iterative rewriting process. Before we motivate our formulation, we start with the description of optimization problems of our interest. Typically, an optimization problem can be defined as follows:

**Definition 1 (Optimization problem.)** *Let $\mathcal{S}$ be the space of all valid states in the problem domain, and $c : \mathcal{S} \to \mathcal{R}$ is the cost function. The goal is to find $\arg\min_{s \in \mathcal{S}} c(s)$.*

When $\mathcal{S}$ is continuous, gradient-based algorithms are effective ways of finding the optimal solutions. For example, trajectory optimization techniques such as iLQR leverage the gradient information to gradually improve the current trajectory towards the optimum (MAYNE, 1973). In contrast, for discrete $\mathcal{S}$, it is not straightforward to apply similar approaches. A long line of works propose algorithms to construct from scratch the optimal solution for combinatorial optimization problems and job scheduling problems (Karp, 1972; Grandl et al., 2015). However, when dealing with more complicated problems, it is challenging to output an effective solution from the ground up.

In comparison, for many optimization problems, a better alternative approach is to first construct a feasible solution, then make incremental improvement. This is because (1) a feasible solution is often easy to find; (2) an existing solution provides a full context for the improvement, which is not the case if a solution is generated from scratch, (3) many problems, as well as their solutions, have strong local structures that can be utilized when improving incrementally, and (4) different solutions might share a common routine towards the optimal.

For example, in job scheduling, it is usually difficult to decide whether a job with a large resource requirement needs to be postponed; however, from an existing schedule, it is easy to identify what jobs cause a long waiting time for later ones. In this case, re-scheduling them improves the efficiency.

Therefore, we propose to solve optimization problems through *rewriting*, as defined formally below.

**Definition 2 (Optimization as a rewriting problem.)** *Let $\mathcal{S}$ be the space of all valid states, $\mathcal{A}$ be the rewriting ruleset, $c : \mathcal{S} \to \mathcal{R}$ is the cost function. Suppose $s_t$ is the state at rewriting iteration $t$, we can apply a rewriting rule $a_t \in \mathcal{A}$ to $\hat{g}_t$, where $\hat{g}_t \subset s_t$, and the rewriting step results in the next state $s_{t+1} = f(s_t, \hat{g}_t, a_t)$. Given an initial state $s_0$, our goal is to find a sequence of rewriting steps $(s_0, (g_0, a_0)), (s_1, (g_1, a_1)), ..., (s_{T-1}, (g_{T-1}, a_{T-1})), s_T$ that minimizes $c(s_T)$.*

To tackle a rewriting problem, rule-based rewriters with manually-designed rewriting routines have been proposed (Halide, 2018). However, manually designing such routines is not a trivial task. An incomplete set of routines often leads to an inefficient exhaustive search, while a set of kaleidoscopic routines is often cumbersome to design, hard to maintain and lacks flexibility.

In this paper, we propose to train a neural network instead, using reinforcement learning. Recent advance in deep reinforcement learning suggests the potential of well-trained models to discover novel effective policies, such as demonstrated in Computer Go (Silver et al., 2017) and video games (OpenAI, 2018). In our evaluation, we demonstrate that our approach not only mitigates laborious human efforts, but also enables the model to discover novel rewriting paths from its own exploration.

In the following sections, we discuss the application of our rewriting approach to two different domains: job scheduling (as mentioned above) and expression simplification, in which we minimize the expression length using a well-defined semantics-preserving rewriting ruleset.

## 3.1 JOB SCHEDULING PROBLEM

We first study the job scheduling problem, and in particular, we consider a simplified problem setup studied in (Mao et al., 2016) as follows.

Suppose we have a machine with $D$ types of resources. Each job $j$ is specified as $g_j = (r_{j1}, r_{j2}, ..., r_{jD}, S_j, T_j)$, where $r_{j1}, r_{j2}, ..., r_{jD}$ denotes the required portion (between $0$ and $1$) of resources for each type, $S_j$ is the arrival timestep, and $T_j$ is the duration. We assume that the resource requirement is fixed during the entire job execution, each job must run continuously until finishing, and no preemption is allowed. We adopt an online setting: there is a pending job queue $Q$ which can hold at most $L_Q$ jobs. When a new job arrives, it can either be allocated immediately, or be added to $Q$. If $Q$ is already full, to make space for the new job, at least one job in the $Q$ needs to be scheduled immediately. The goal is to find a time schedule for every job, so that the average waiting time is as short as possible.

For job scheduling, the only type of rewriting is to re-schedule a job $g_j$ and allocate it after another job $g_{j'}$ finishes or at its arrival time $S_j$. Details of a rewriting step is presented in Appendix B.1. Thus, the size of the rewriting ruleset is $|\mathcal{A}| \models 2L_Q$, since each job could only switch its scheduling order with at most $L_Q$ of its former and latter jobs respectively.

## 3.2 EXPRESSION SIMPLIFICATION

We also apply our approach to expression simplification domain. In particular, we consider expressions in Halide, a domain-specific language for high-performance image processing (Ragan-Kelley et al., 2013), which is widely used at scale in multiple products of Google (e.g., YouTube) and Adobe Photoshop. Simplifying Halide expressions is an important step towards the optimization of the entire code. To this end, a rule-based rewriter is implemented for the expressions, which is carefully tuned with manually-designed heuristics [1]. The grammar of the expressions considered in the rewriter is specified in Appendix A. Notice that the grammar includes a more comprehensive operator set than previous works on finding equivalent expressions, which consider only boolean expressions (Allamanis et al., 2017; Evans et al., 2018) or a subset of algorithmic operations (Allamanis et al., 2017). The rewriter includes hundreds of manually-designed rewriting templates. Given an expression, the rewriter checks the templates in a pre-designed order, and applies those rewriting templates that match any sub-expression of the input.

After investigating into the rewriting templates in the rule-based rewriter, we find that a large number of rewriting templates enumerate specific cases for an *uphill rule*, which lengthens the expression first and shortens it later (e.g., "min/max" expansion). Similar to momentum terms in gradient descent for continuous optimization, such rules are used to escape a local optimum. However, they should only be applied when the initial expression satisfies certain *pre-conditions*, which is traditionally specified by manual design, a cumbersome process that is hard to generalize.

Observing these limitations, we hypothesize that a neural network model has the potential of doing a better job than the rule-based rewriter. In particular, we propose to only keep the core rewriting rules in the ruleset, remove all unnecessary pre-conditions, and let the neural network decide which and when to apply each rewriting rule. In this way, the neural rewriter has a better flexibility than the rule-based rewriter, because it can learn such rewriting decisions from data, and has the ability of discovering novel rewriting patterns that are not included in the rule-based rewriter.

## 4 NEURAL REWRITER MODEL

---

[1] The code is released in their public repository here: `https://github.com/halide/Halide`.

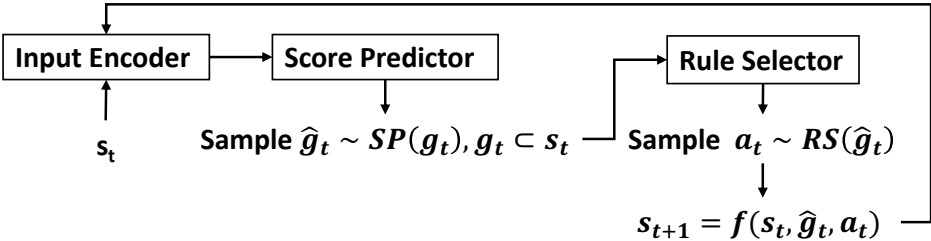

Figure 1: The framework of the neural rewriter architecture. Here, the score predictor computes $SP(g_t)$, which is the rewriting score; and the rule selector predicts $RS(\hat{g}_t)$, which is the probability distribution of applying each rewriting rule $a \in \mathcal{A}$.

In the following, we present the design of our rewriting model, i.e., *Neural Rewriter*. We first provide an overview of our model framework, then present the design details for different applications.

## 4.1 Model Overview

Figure 1 illustrates the overall framework of our neural rewriter, and we describe the two key components for rewriting as follows. More details can be found in Appendix C.

**Score predictor.** Given the state $s_t$, the score predictor computes a score $SP(g_t)$ for every $g_t \subset s_t$, which measures the benefit of rewriting $g_t$. A high score indicates that rewriting $g_t$ could be desirable.

**Rule selector.** Given $\hat{g}_t \subset s_t$ to be rewritten, the rule selector predicts a probability distribution $RS(\hat{g}_t)$ over the entire ruleset $\mathcal{A}$, and selects a rule $a_t \in \mathcal{A}$ to apply accordingly.

## 4.2 Model Details for Job Scheduling Problem

Figure 2a demonstrates the model architecture for job scheduling, and we discuss the details below.

**Input embedding.** As described in Section 3, each job is specified by $g_j = (r_{j1}, r_{j2}, ..., r_{jD}, S_j, T_j)$. In addition, we define $A_j$ as the schedule time, and $C_j = A_j + T_j$ as the completion time. We embed each job into a $(D \times (T_{max} + 1) + 1)$-dimensional vector $e_j$, where $T_{max}$ is the maximal duration of a job. This vector encodes the information of its attributes and the machine status during its execution, and we defer the embedding details to Appendix C.1.

We represent each schedule as a directed acyclic graph (DAG), which describes the dependency among the schedule time of different jobs. Specifically, we denote each job $g_j$ as a node in the graph, and we add an additional node $g_0$ to represent the machine, which has a zero embedding vector $e_0$. If a job $g_j$ is scheduled at its arrival time $S_j$, then we add a directed edge $\langle g_0, g_j \rangle$ in the graph. Otherwise, there must exist at least one job $g_{j'}$ such that $C_{j'} = A_j$. We add an edge $\langle g_{j'}, g_j \rangle$ for every such job $g_{j'}$ to the graph. Figure 3 illustrates an example of the graph construction.

To encode the graphs, we extend the Child-Sum Tree-LSTM architecture in (Tai et al., 2015), which is similar to the DAG-structured LSTM in (Zhu et al., 2016). Specifically, for a job $g_j$, suppose $(h_1, c_1), (h_2, c_2), ..., (h_p, c_p)$ are the LSTM states of all parents of $g_j$, then its LSTM state is

$$(h, c) = \text{LSTM}((\sum_{i=1}^{p} h_i, \sum_{i=1}^{p} c_i), e_j) \tag{1}$$

For each node, the $d$-dimensional hidden state $h$ is used as the embedding for other two components.

**Score predictor.** This component is a $L_P$-layer fully connected neural network with a hidden size of $N_p$, and the input to the predictor of job $g_j$ is $h_j$.

**Rule selector.** The rule selector is a $L_S$-layer fully connected neural network with a hidden size of $N_S$. The rewriting options described in Section 3 is equivalent to moving the current job $g_j$ to be a child of another job $g_{j'}$ or $g_0$ in the graph, which means allocating job $g_j$ after job $g_{j'}$ finishes or at its arrival time $S_j$. Thus, the input to the rule selector not only includes $h_j$, but also $h_{j'}$ of all other

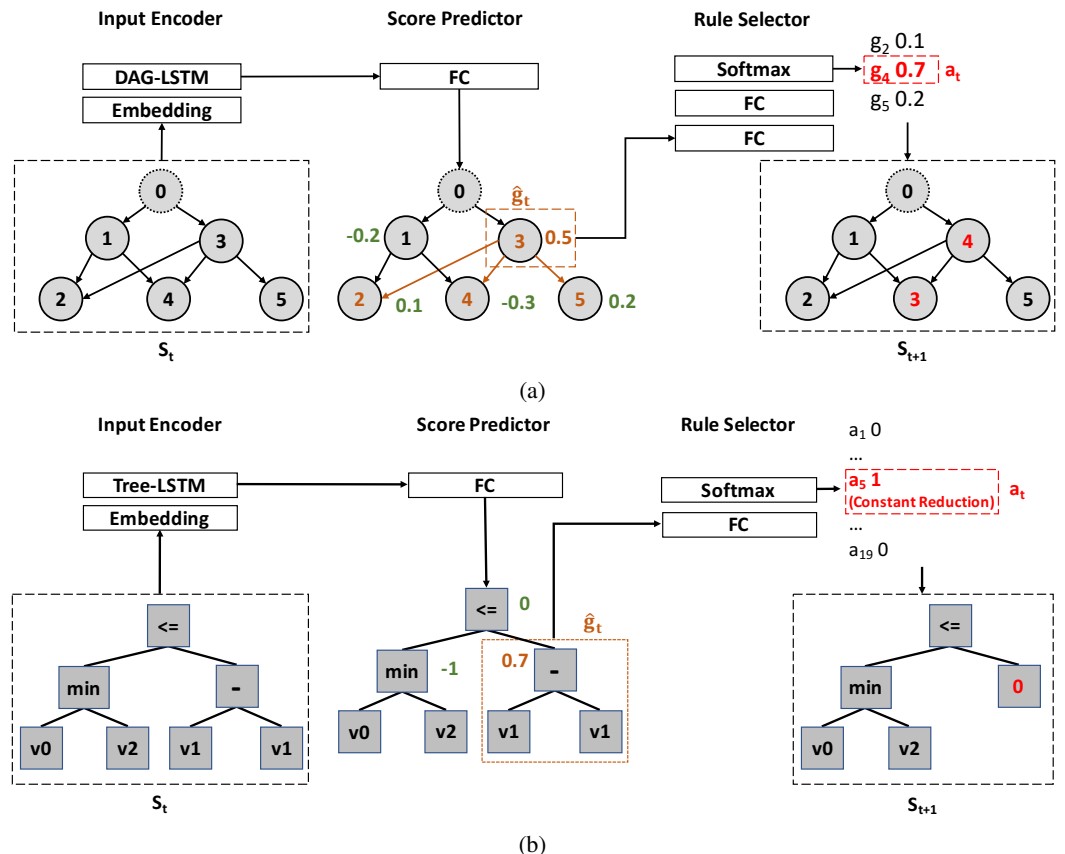

Figure 2: The instantiation of neural rewriter architectures for different domains: **(a)** job scheduling; **(b)** expression simplification. In **(a)**, $s_t$ is the dependency graph representation of the job schedule. Each circle with index greater than 0 represents a job node, and node 0 is an additional one representing the machine. Edges in the graph reflect job dependencies. The score predictor selects a job $\hat{g}_t \sim SP(g_t)$ from all job nodes to re-schedule. The rule selector chooses a moving action $a_t \in \mathcal{A}$ for $\hat{g}_t$, then modifies $s_t$ to get a new dependency graph $s_{t+1}$. In **(b)**, $s_t$ is the expression parse tree, where each square represents a node in the tree. The set of $g_t \subset s_t$ includes every sub-tree rooted at a non-terminal node, from which the score predictor selects $\hat{g}_t$ to rewrite. Afterwards, the rule selector predicts a rewriting rule $a_t$, then rewrites the sub-tree $\hat{g}_t$ to get the new tree $s_{t+1}$.

$g_{j'}$ that could be used for rewriting. The output layer is an $|\mathcal{A}|$-dimensional softmax layer, where $|\mathcal{A}| = 2L_Q$ as discussed in Section 3. More details can be found in Appendix C.1.

### 4.3 MODEL DETAILS FOR EXPRESSION SIMPLIFICATION

We present the instantiation of our neural rewriter framework for expression simplification in Figure 2b. We mainly discuss the design choices different from the model for job scheduling below.

**Input embedding.** We use expression parse trees as the input, and employ the N-ary Tree-LSTM designed in (Tai et al., 2015) as the input encoder to compute the embedding for each node in the tree. Notice that in this problem, each non-terminal has at most 3 children. Thus, let $x$ be the embedding of a non-terminal, $(h_L, c_L), (h_M, c_M), (h_R, c_R)$ be the LSTM states maintained by its children nodes, the LSTM state of the non-terminal node is computed as

$$(h, c) = \text{LSTM}(([h_L; h_M; h_R], [c_L; c_M; c_R]), x) \qquad (2)$$

Where $[a; b]$ denotes the concatenation of vectors $a$ and $b$. For non-terminals with less than 3 children, the corresponding LSTM states are set to be zero.

**Score predictor.** The score predictor is a $L_P$-layer fully connected neural network with a hidden size of $N_P$. For each sub-tree $g_i$, its input to the score predictor is represented as a $2d$-dimensional vector $[h_0; h_i]$, where $h_0$ embeds the entire tree. More details can be found in Appendix C.2.

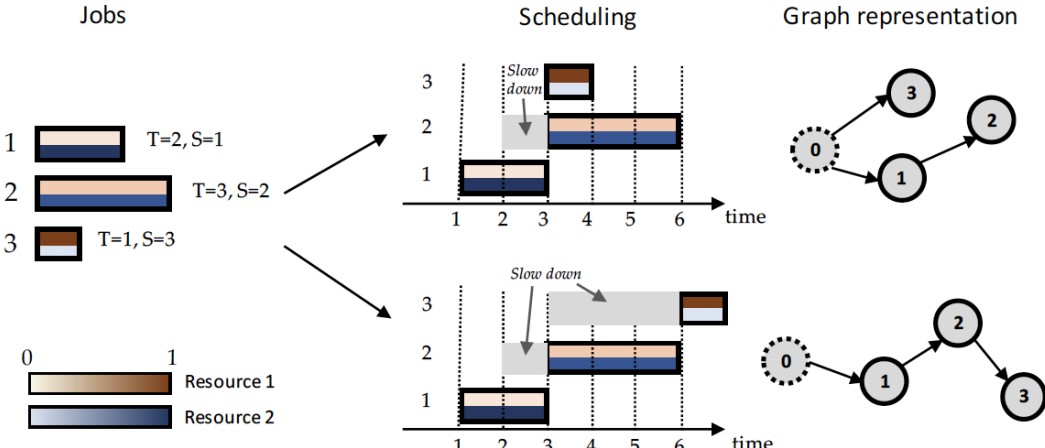

Figure 3: An example to illustrate the graph construction approach for the job scheduling problem. Node 0 is an additional node representing the machine.

**Rule selector.** The rule selector is a $L_S$-layer fully connected neural network with a hidden size of $N_S$, and its input format is the same as the score predictor.

## 4.4 TRAINING DETAILS

Let $(s_0, (\hat{g}_0, a_0)), (s_1, (\hat{g}_1, a_1)), ..., (s_{T-1}, (\hat{g}_{T-1}, a_{T-1})), s_T$ be the rewriting sequence in the forward pass, we define the reward function $r(s_t, (\hat{g}_t, a_t))$ as follows. For job scheduling problem, $r(s_t, (\hat{g}_t, a_t)) = c(s_t) - c(s_{t+1})$. For expression simplification problem, $r(s_t, (\hat{g}_t, a_t)) = \max_{t+1 \leq k \leq T} (\gamma^{k-t}(c(s_t) - c(s_k)))/c(\hat{g}_t)$, where $\gamma$ is a discount factor. This reward function is designed to credit uphill rules that could lead to a simplified expression in the end. More discussion about the forward pass algorithm and reward design can be found in Appendix D.

The reward function $r(s_t, (\hat{g}_t, a_t))$ is also used as the training target of $SP(\hat{g}_t)$. Specifically, let $\theta$ be the model parameters, then the loss function of the score predictor is

$$L_{SP}(\theta) = \frac{1}{T} \sum_{t=0}^{T-1} (r(s_t, (\hat{g}_t, a_t)) - SP(\hat{g}_t))^2 \tag{3}$$

To train the rule selector, we employ the Advantage Actor-Critic algorithm (Sutton et al., 1998), and we use $SP(\hat{g}_t)$ as the critic. In particular, let the advantage function $A(s_t, (\hat{g}_t, a_t)) = r(s_t, (\hat{g}_t, a_t)) - SP(\hat{g}_t)$, then the loss function of the rule selector is

$$L_{RS}(\theta) = - \sum_{t=0}^{T-1} A(s_t, (\hat{g}_t, a_t)) \log RS(\hat{g}_t) \tag{4}$$

The overall loss function is $L(\theta) = L_{RS}(\theta) + \alpha L_{SP}(\theta)$, where $\alpha$ is a hyper-parameter.

## 5 EXPERIMENTS

In this section, we present the evaluation results of both job scheduling and expression simplification problems. All neural network models in our evaluation are implemented in PyTorch (Paszke et al., 2017). To calculate the inference time during testing, we run all algorithms on the same server equipped with 2 Quadro GP100 GPUs and 80 CPU cores. Only 1 GPU is used when evaluating neural network models, and 4 CPU cores are used for search algorithms . For both tasks, we set the timeout of search algorithms to be 10 times as long as the timeout of our neural rewriter.

| D | 2 | 5 | 10 | 20 |
|---|---|---|---|---|
| Shortest Job First | 4.80 | 5.83 | 5.58 | 5.00 |
| Shortest First Search | 4.25 | 5.05 | 5.54 | 4.98 |
| DeepRM (Mao et al., 2016) | 2.81 | 6.52 | 9.20 | 10.18 |
| Neural Rewriter (Ours) | **2.80** | **3.36** | **4.50** | **4.63** |
| Optim (Lower bound) | 2.57 | 3.02 | 4.08 | 4.26 |
| Earliest Job First (Upper bound) | 11.11 | 13.62 | 22.13 | 24.23 |

Table 1: Experimental results of the job scheduling problem on the test set. For each approach, we report the average slowdown of the jobs with different number of resource types $D$.

## 5.1 JOB SCHEDULING PROBLEM

### 5.1.1 EVALUATION SETUP

We randomly generate $100K$ job sequences, and use $80K$ for training, $10K$ for validation, and $10K$ for testing. We use an online setting where jobs arrive on the fly with a pending job queue of length $L_Q = 10$. When the number of resource types $D = 2$, we follow the same setup as in (Mao et al., 2016). The maximal job duration $T_{max} = 15$, and the latest job arrival time is $S_{max} = 50$. With larger $D$, except changing the resource requirement of each job to include more resource types, other configurations stay the same. We use *average job slowdown* as our evaluation metric, which is computed by $(C_j - S_j)/T_j$. This metric is also used in (Mao et al., 2016).

### 5.1.2 MODEL CONFIGURATION

For our neural rewriter model, we provide an initial schedule that allocates jobs in a first-come-first-serve manner for every job sequence, then feed it to the neural rewriter for refinement. Such an initial schedule is intuitive to design, easy to compute with a negligible overhead, while is much less effective than the optimal solution, as we will demonstrate in Table 1.

We compare our neural rewriter model with two kinds of baselines. The first kind of baselines use manually designed heuristics: *Shortest Job First (SJF)* always allocates the shortest job in the pending job queue at each timestep, also used as a baseline in (Mao et al., 2016). *Shortest First Search* searches over the shortest $k$ jobs to schedule at each timestep, and returns the optimal one. We find that other heuristic-based baselines used in (Mao et al., 2016) generally perform worse than *SJF*, especially with large $D$. Thus, we omit the comparison.

The second kind of baselines use a deep neural network to construct the job schedule from scratch. We re-implement *DeepRM* (Mao et al., 2016), a neural network trained with reinforcement learning, and test it on larger $D$. For a fair comparison, we tune the hyper-parameters of *DeepRM* for best performance.

To measure the optimality of these algorithms, we also compute the following empirical bounds. *Earliest Job First (EJF)* schedules each job by their arrival time. This provides an upper bound of the average job slowdown. We also use this algorithm to generate initial schedules for the neural rewriter. *Optim* assumes an offline setting: the entire job sequence is available before scheduling. This leads to a better algorithm that sorts jobs' duration at the earliest time runnable on the machine. It gives a lower bound of the average job slowdown in an idealized setting.

### 5.1.3 RESULTS

Table 1 presents the results for the job scheduling problem. Our neural rewriter model outperforms both heuristic algorithms and the baseline neural network *DeepRM*. In particular, we observe that while the performance of *DeepRM* and our neural rewriter are similar when $D = 2$, with larger $D$, *DeepRM* starts to perform worse than heuristic-based algorithms, which is consistent with our hypothesis that it becomes challenging to design a schedule from scratch when the environment becomes more complex. On the other hand, our neural rewriter could capture the bottleneck of an existing schedule that limits its efficiency, then progressively refine it to obtain a better one. Meanwhile, our results are also closer to the empirical lower bound computed by the *Optim* algorithm, which further demonstrates the effectiveness of our rewriting approach. More results and discussion can be found in Appendix E.

| Number of expressions in the dataset | Length of expressions | Size of expression parse trees |
|---|---|---|
| Total: 1.36M | Average: 106.84 | Average: 27.39 |
| Training/Val/Test: 1.09M/136K/136K | Min/Max: 10/579 | Min/Max:3/100 |

Table 2: Statistics of the dataset for expression simplification.

| | Average expression length reduction | Average tree size reduction |
|---|---|---|
| Halide Rule-based Rewriter | 36.13 | 9.68 |
| Heuristic Search | 43.27 | 12.09 |
| Neural Rewriter (Ours) | **46.98** | **13.53** |
| Z3 Solver | 50.81 | 15.82 |

Table 3: Experimental results of the Halide expression simplification task on the test set. Note that the Z3 solver can perform rewriting steps that are not included in the Halide ruleset.

## 5.2 EXPRESSION SIMPLIFICATION

### 5.2.1 EVALUATION SETUP

To construct the dataset, we first generate random pipelines using the generator in the Halide repository [2], then extract expressions from them. We filter out those expressions that can not be further rewritten, then split the rest into 8/1/1 for training/validation/test set respectively. We summarize the statistics of the dataset in Table 2, and more details can be found in Appendix A.

We measure the following metrics in our evaluation: (1) *Average expression length reduction*, which is the length reduced from the initial expression to the rewritten one, and the length is defined as the number of characters in the expression; (2) *Average tree size reduction*, which is the number of nodes decreased from the initial expression parse tree to the rewritten one.

### 5.2.2 MODEL CONFIGURATION

We discuss the ruleset design of our neural rewriter as follows. We look into the Halide rewriting ruleset with hundreds of templates, and for those templates that can not be further simplified, e.g., reducing $v - v$ into $0$ as in Figure 2b, we simply include them in our ruleset. As discussed in Section 3, a large number of templates are enumerating pre-conditions to apply uphill rules. For these templates, we remove the manually-designed pre-conditions, and only include the uphill rules themselves in the ruleset. In this way, we manually build a ruleset with $|\mathcal{A}| \models 19$ categories of rewriting rules. More details about the ruleset and rewriting process can be found in Appendix B.2.

We examine the effectiveness of our neural rewriter against two kinds of baselines. The first kind of baselines are heuristic-based rewriting approaches, including the rule-based Halide rewriter (Section 3) and a heuristic search algorithm using our ruleset only. In each rewriting iteration, the search algorithm first enumerates all combinations of rewriting node and rule, then selects the top $k$ shortest resulted expressions to be rewritten in the next iteration.

In addition, we evaluate Z3 solver on our dataset (De Moura & Bjørner, 2008). Z3 solver is a high-performance theorem prover developed by Microsoft Research. Its simplifier works by traversing each sub-formula in the input expression and invoking the solver to find a simpler equivalent one to replace it. Therefore, the simplification steps performed by this solver may not be included in the Halide ruleset, which makes it a strong baseline to compare with. We set the timeout to be 10 seconds for each input expression, and we find that the results are not significantly better with a longer timeout.

### 5.2.3 RESULTS

Table 3 presents the main results of our expression simplification problem. We can observe that our neural rewriter model outperforms both the rule-based rewriter and the heuristic search by a large margin. In particular, our neural rewriter could reduce the expression length and parse tree size by around $45\%$ on average; meanwhile, compared to the rule-based rewriter, our model further reduces the average expression length and tree size by $30\%$ and $40\%$ respectively. We observe that the main performance gain comes from learning to apply uphill rules appropriately in ways that are not included in the manually-designed templates. For example, consider the

---

[2] `https://github.com/halide/Halide/tree/new_autoschedule_with_new_ simplifier/apps/random_pipeline.`

expression $5 \leq \max(\max(v0, 3) + 3, \max(v1, v2))$, which could be reduced to $True$ by expanding $\max(\max(v0, 3) + 3, \max(v1, v2))$ and $\max(v0, 3)$. Using a rule-based rewriter would require the need of specifying the pre-conditions recursively, which becomes prohibitive when the expressions become more complex. On the other hand, heuristic search may not be able to find the correct order of expanding the right hand size of the expression when more "min/max" are included, which would make the search less efficient. Meanwhile, the performance of our neural rewriter is also much closer to the performance of Z3 solver, which could perform rewriting steps that are not included in the Halide ruleset. More results can be found in Appendix F.

## 6 CONCLUSION

In this work, we propose to formulate optimization as a rewriting problem, and solve the problem by iteratively rewriting an existing solution towards the optimum. We utilize deep reinforcement learning to train our neural rewriter. In our evaluation, we demonstrate the effectiveness of our neural rewriter on job scheduling and expression simplification problems, where our model outperforms heuristic-based algorithms and baseline deep neural networks that generate an entire solution directly.

Meanwhile, we observe that since our approach is based on local rewriting, it could become time-consuming when large changes are needed. In extreme cases where each rewriting step needs to change the global structure, starting from scratch becomes preferrable. We consider improving the efficiency of our rewriting approach and extending it to more complicated scenarios as future work.

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

```
       <Expr>    ::=   <AlgExpr> | <BoolExpr>
    <BoolExpr>   ::=   <AlgExpr> << <AlgExpr>
                  |    <AlgExpr> <= <AlgExpr>
                  |    <AlgExpr> == <AlgExpr>
                  |    (!<BoolExpr>)
                  |    (<BoolExpr> && <BoolExpr>)
                  |    (<BoolExpr> || <BoolExpr>)
     <AlgExpr>   ::=   <Term>
                  |    (<AlgExpr> + <Term>)
                  |    (<AlgExpr> - <Term>)
                  |    (<AlgExpr> * <Term>)
                  |    (<AlgExpr> / <Term>)
                  |    (<AlgExpr> % <Term>)
       <Term>    ::=    | <Const>
                  |    max(<AlgExpr>, <AlgExpr>)
                  |    min(<AlgExpr>, <AlgExpr>)
                  |    select(<BoolExpr>, <AlgExpr>, <AlgExpr>)
```

Figure 4: Grammar of the Halide expressions in our evaluation. "select ($c$, $e1$, $e2$)" means that when the condition $c$ is satisfied, this term is equal to $e1$, otherwise is equal to $e2$. In our dataset, all constants are integers ranging in $[-1024, 1024]$, and variables are from the set $\{v0, v1, ..., v12\}$.

---

**Algorithm 1** Algorithm of a Single Rewriting Step for Job Scheduling Problem

---

1: **function** REWRITE($g_j, g_{j'}, s_t$)
2:     **if** $C_{j'} < S_j$ or $C_{j'} == A_j$ **then**
3:         **return** S
4:     **end if**
5:     **if** $j' \neq 0$ **then** $A'_j = C_{j'}$ **else** $A'_j = S_j$ **fi**
6:     $C'_j = A'_j + T_j$
7:
8:     //Resolve potential resource occupation overflow within $[A'_j, C'_j]$
9:     $J$ = all jobs in $s_t$ except $g_j$ that are scheduled within $[A'_j, C'_j]$
10:     Sort $J$ in the topological order
11:     **for** $g_i \in J$ **do**
12:         $A'_i$ = the earliest time that job $g_i$ can be scheduled
13:         $C'_i = A'_i + T_i$
14:     **end for**
15:     For $g_i \notin J$, $A'_i = A_i$, $C'_i = C_i$
16:     $s_{t+1} = \{(A'_i, C'_i)\}$
17:     **return** $s_{t+1}$
18: **end function**

---

## A  GRAMMAR OF THE HALIDE EXPRESSIONS

Figure 4 presents the grammar of Halide expressions in our evaluation.

## B  MORE DETAILS ON THE REWRITING RULESET

### B.1  MORE DETAILS FOR JOB SCHEDULING PROBLEM

Algorithm 1 describes a single rewriting step for job scheduling problem.

### B.2  MORE DETAILS FOR EXPRESSION SIMPLIFICATION PROBLEM

**More discussions about the uphill rules.** A commonly used type of uphill rules is "min/max" expansion, e.g., $\min(a, b) < c \rightarrow a < c || b < c$. Dozens of templates in the ruleset of the Halide rewriter are describing conditions when a "min/max" expression could be simplified. Notice that although applying this rewriting rule has no benefit in most cases, since it will increase the expression

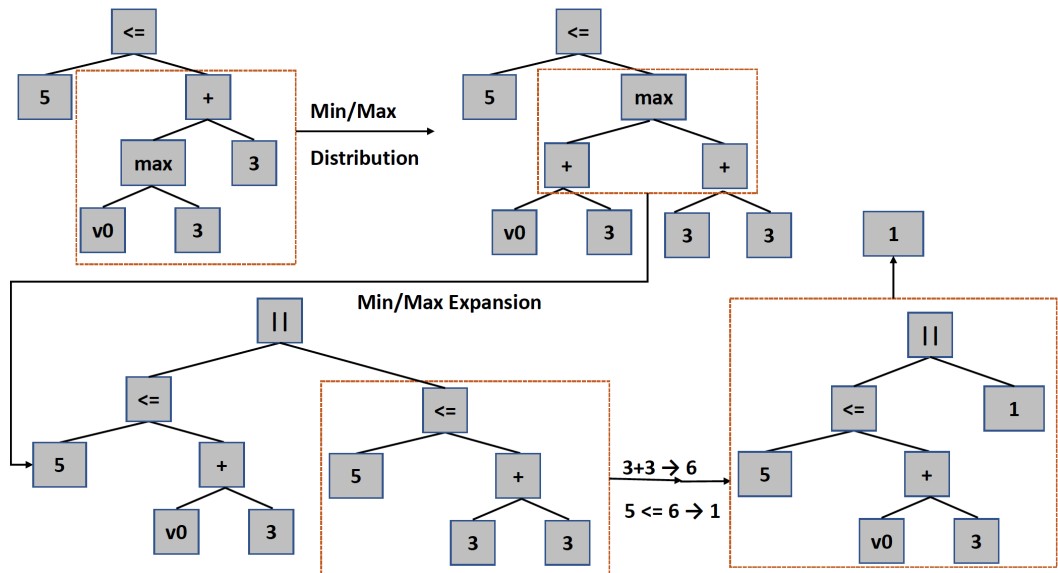

Figure 5: An example of the rewriting process for Halide expressions. The initial expression is $5 \leq max(v0, 3) + 3$, which could be reduced to 1, i.e., $True$.

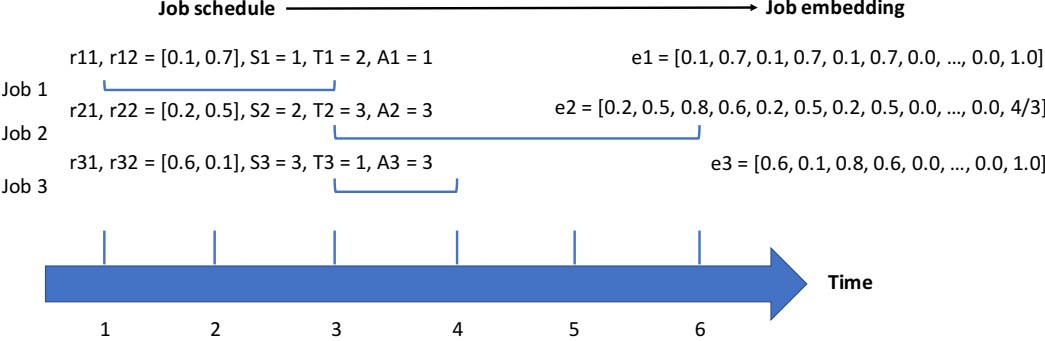

Figure 6: An example to illustrate the job embedding approach for the job scheduling problem.

length, it is necessary to include it in the ruleset, because when either $a < c$ or $b < c$ is always true, expanding the "min" term could reduce the entire expression to a tautology, which ends up simplifying the entire expression. Figure 5 shows an example of the rewriting process using uphill rules properly.

## C  MORE DETAILS ON MODEL ARCHITECTURES

### C.1  MORE DETAILS FOR JOB SCHEDULING PROBLEM

**Job embedding.**  We describe the details of job embedding as follows. Consider a job $g_j = (r_{j1}, r_{j2}, ..., r_{jD}, S_j, T_j)$. We denote the amount of resources occupied by all jobs at each timestep $t$ as $r'_t = (r'_{t1}, r'_{t2}, ..., r'_{tD})$. Each job $g_j$ is represented as a $(D \times (T_{max} + 1) + 1)$-dimensional vector, where the first $D$ dimensions of the vector are $(r_{j1}, r_{j2}, ..., r_{jD})$, representing its resource requirement. The following $D \times T_j$ dimensions of the vector are the concatenation of $r'_{A_j}, r'_{A_j+1}, ..., r'_{A_j+T_j-1}$, which describes the machine usage during the execution of the job $g_j$. When $T_j < T_{max}$, the following $D \times (T_{max} - T_j)$ dimensions are zero. The last dimension of the embedding vector is the *slowdown* of the job in current schedule. The definition of the slowdown is the same as in (Mao et al., 2016), which is computed by $(C_j - S_j)/T_j$, where $C_j = A_j + T_j$ is the completion time. We denote the embedding of each job $g_j$ as $e_j$. Figure 6 shows an example of our job embedding approach.

---

**Algorithm 2** Forward Pass Algorithm for the Neural Rewriter during Training

---

**Require:** initial state $s_0$, hyper-parameters $\beta, \epsilon, p_c, T_{iter}, T_{sp}, T_{rs}$
1: **for** $t = 0 \rightarrow T_{iter} - 1$ **do**
2:      **for** $i = 1 \rightarrow T_{sp}$ **do**
3:          Sample $\hat{g}_t \sim \exp(\beta \cdot SP(g_t))$, where $g_t \subset s_t$
4:          **if** $SP(\hat{g}_t) < \epsilon$ **then**
5:              Re-sample $\hat{g}'_t \sim \exp(\beta \cdot SP(g_t))$ with a probability of $1 - p_c$
6:              **if** Re-sampling is not performed **then break fi**
7:          **else**
8:              **break**
9:          **end if**
10:      **end for**
11:      **for** $i = 1 \rightarrow T_{rs}$ **do**
12:          Sample $a_t \sim RS(\hat{g}_t)$
13:          **if** $a_t$ can be applied to $(s_t, \hat{g}_t)$ **then break fi**
14:      **end for**
15:      **if** $a_t$ does not applied to $(s_t, \hat{g}_t)$ **then break fi**
16:      $s_{t+1} = f(s_t, \hat{g}_t, a_t)$
17: **end for**

---

**Rule selector.** The rule selector has two modules. The first module is a $L_S$-layer fully connected neural network with a hidden size of $N_S$. For each job $g_j$, let $N_j$ be the number of jobs that could be the parent of $g_j$, and $\{g_{j'_k}\}$ denotes the set of such jobs. For each $g_{j'_k}$, the input is $[h_j; h_{j'_k}]$, and this module computes a $d$-dimensional vector $h'_k$ to encode such a pair of jobs. The second module of the rule selector is another $L_S$-layer fully connected neural network with a hidden size of $N_S$. For this module, the input is an $(|\mathcal{A}| \times d)$-dimensional vector $[h'_1; h'_2; ...; h'_{|\mathcal{A}|}]$, where $|\mathcal{A}| \models 2L_Q$. When $N_j < |\mathcal{A}|$, $h'_{N_j+1}, h'_{N_j+2}, ..., h'_{|\mathcal{A}|}$ are set to be zero. The output layer of this module is an $|\mathcal{A}|$-dimensional softmax layer, which predicts the probability of each different move of $g_j$.

### C.2 MODEL DETAILS FOR EXPRESSION SIMPLIFICATION

**Input representation.** As discussed in Section 4, for each sub-tree $g_i$, its input to both the score predictor and the rule selector is represented as a $2d$-dimensional vector $[h_0; h_i]$, where $h_0$ is the embedding of the root node encoding the entire tree. The reason why we include $h_0$ in the input is that looking at the sub-tree itself is sometimes insufficient to determine whether it is beneficial to perform the rewriting. For example, consider the expression $max(a, b) + 2 < a + 2$, by looking at the sub-expression $max(a, b) + 2$ itself, it does not seem necessary to rewrite it as $max(a+2, b+2)$. However, given the entire expression, we can observe that this rewriting is an important step towards the simplification, since the resulted expression $max(a + 2, b + 2) < a + 2$ could be reduced to $false$. We have tried other approaches of combining the parent information into the input, but we find that including the embedding of the entire tree is the most efficient way.

### C.3 MODEL HYPER-PARAMETERS

For both tasks, $L_S = L_P = 1$, $N_S = N_P = 256$, $d = 512$.

## D MORE DETAILS ON TRAINING

Algorithm 2 presents the details of the forward pass during training. The forward pass during evaluation is similar, except that we compute $\hat{g}_t$ and $a_t$ as $\hat{g}_t = \arg\max_{g_t} SP(g_t)$ and $a_t = \arg\max_a(RS(\hat{g}_t))$, and the inference immediately terminates when $SP(\hat{g}_t) < \epsilon$ or $a_t$ does not apply.

**Reward design.** For job scheduling problem, we simply define the reward function as $r(s_t, (\hat{g}_t, a_t)) = c(s_t) - c(s_{t+1})$. The same reward function does not apply to the expression simplification problem, since it would always give a negative reward for uphill rewriting rules such as expanding a "min/max". Thus, for expression simplification problem, we modify the reward function to $r(s_t, (\hat{g}_t, a_t)) = max_{t+1 \leq k \leq T}(\gamma^{k-t}(c(s_t) - c(s_k)))/c(\hat{g}_t)$, where $\gamma$ is a discount factor. This

| Initial average slow down | $\leq 10$ | $10 - 25$ | $> 25$ |
|---|---|---|---|
| Final average slow down | 4.49 | 4.61 | 4.78 |
| Results from Table 1 | | | |
| Shortest Job First | 5.00 | | |
| Shortest First Search | 4.98 | | |
| DeepRM (Mao et al., 2016) | 10.18 | | |
| Neural Rewriter (Ours) | **4.63** | | |
| Optim (Lower bound) | 4.26 | | |
| Earliest Job First (Upper bound) | 24.23 | | |

Table 4: Experimental results of the job scheduling problem using initial schedules with different average slow down. The number of resource types $D = 20$. We also include the results from Table 1 for reference.

design of reward function would assign a positive value to an uphill rule if it is included in a path that results in a simplified expression in the end. We normalize the reward by $c(\hat{g}_t)$, so that the reward function is bounded. In our evaluation, we set $\gamma = 0.9$.

**Hyper-parameters.** In Algorithm 2, $\beta = 10.0, \epsilon = 0.0, T_{sp} = 10, T_{rs} = 10, T_{iter} = 50$. $p_c$ is initialized with $0.5$, and is decayed by $0.8$ for every 1000 timesteps until $p_c = 0.01$, where it is not decayed anymore. In the training loss function, $\alpha = 10.0$. The initial learning rate is $1e - 4$, and is decayed by $0.9$ for every 1000 timesteps. Batch size is 128. Gradients with $L_2$ norm larger than $5.0$ are scaled down to have the norm of $5.0$. The model is trained using Adam optimizer. All weights are initialized uniformly randomly in $[-0.1, 0.1]$.

## E  MORE RESULTS FOR JOB SCHEDULING PROBLEM

To examine how the initial schedules affect the final results, besides earliest-job-first schedules discussed in Section 5, we also evaluate initial schedules with different average slow down. Specifically, for each job sequence, we generate different initial schedules by randomly allocating one job at a time.

In Table 4, we present the results with $D = 20$ types of resources. For each job sequence, we randomly generate 10 different initial schedules. We can observe that although the effectiveness of initial schedules affect the final schedules, the performance is consistently better than other baseline approaches, which demonstrates that our neural rewriter is able to substantially improve the initial solution regardless of its quality.

## F  MORE RESULTS FOR EXPRESSION SIMPLIFICATION

In Figures 7 and 8, we present some success cases of expression simplification, where we can simplify better than the Halide rule-based rewriter.

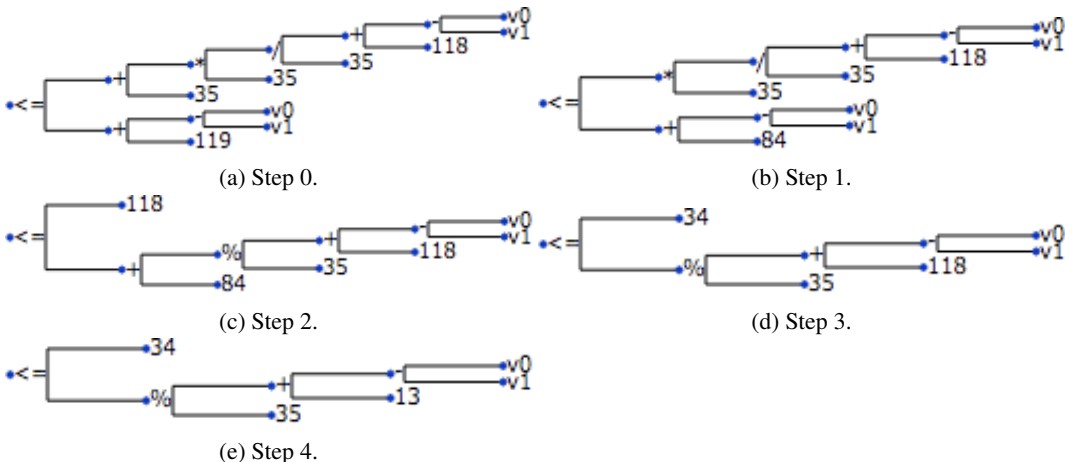

(a) Step 0.      (b) Step 1.

(c) Step 2.      (d) Step 3.

(e) Step 4.

Figure 7: The rewriting process that simplifies the expression $((v0 - v1 + 18)/35 * 35 + 35) \leq v0 - v1 + 119$ to $34 \leq (v0 - v1 + 13)\%35$.

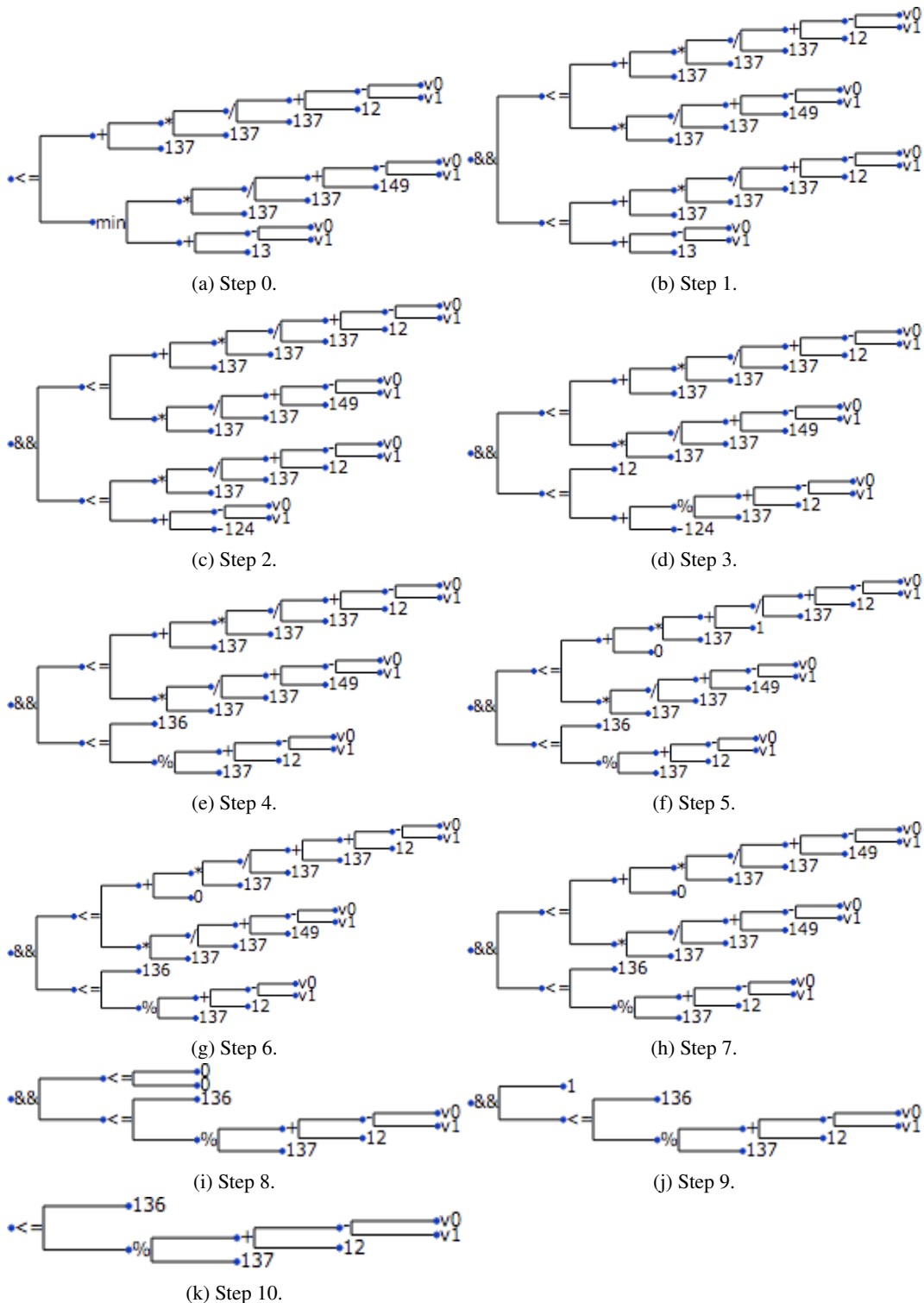

Figure 8: The rewriting process that simplifies the expression $((v0 - v1 + 12)/137 * 137 + 137) \leq min((v0 - v1 + 149)/137 * 137, v0 - v1 + 13)$ to $136 \leq (v0 - v1 + 12)\%137$.

