# OpenReview forum: "Learning to Progressively Plan"
_ICLR.cc/2019/Conference_

### Official Review · AnonReviewer3 · 2018-10-30
**An application of tree and DAG LSTMs with important details missing from the draft**

**Rating:** 5
**Confidence:** 3

**Review:**

The paper proposes to plan by taking an initial plan and improving it. The authors claim that 1) this will achieve results faster than planning from scratch and 2) will lead to better results than using quick, local heuristics. However, when starting with an initial solution there is always the danger of the final solution being overly biased by the initial solution. The authors do not address this adequately. They show how to apply tree and DAG-based LSTMs to job scheduling and shortening expressions. Since they are simply using previously proposed LSTM variants, I do not see much contribution here. The experiments show some gains on randomly generated datasets. More importantly, details are missing such as the definitions of SP and RS from section 4.4.

---

> ### Author Response · Authors · 2018-11-11
> **Response and clarification**
>
> Thank you for your review! About your comments:
>
> - “However, when starting with an initial solution there is always the danger of the final solution being overly biased by the initial solution.”
>
> The motivation of our approach is to improve from existing solutions, and we agree that the rewriting process varies with different initial solutions, e.g., a nearly optimal solution would require a much fewer rewriting steps. However, we note that the quality of the final solution does not heavily depend on the initial one. In fact, we did experiments on job scheduling tasks with random initial schedules, and we found that the neural rewriter model achieves similar performance to the results starting with the earliest job first schedules, as reported in the paper (Section 5.1 on page 7). This demonstrates that our rewriting model is not overly biased by the initial solution. We will perform an ablation study about this point in our revision.
>
> - “Since they are simply using previously proposed LSTM variants, I do not see much contribution here.”
>
> We do not claim that each individual component of our model is novel; instead, our key contribution is the overall framework (Fig. 1) that learns a neural network to progressively improve existing planning in the discrete space, and training the framework with reinforcement learning.
>
> - “More importantly, details are missing such as the definitions of SP and RS from section 4.4.”
>
> These definitions are in Section 4.1 (page 4).

---

### Official Review · AnonReviewer1 · 2018-11-03
**Seems novel, but the evaluations could use some work**

**Rating:** 5
**Confidence:** 3

**Review:**


Summary:
Search-based policies are stronger than a reactive policies, but the resulting time consumption can be exponential. Existing solutions include designing a plan from scratch given a complete problem specification or performing iterative rewriting of the plan, though the latter approach has only been explored in problems where the action and state spaces are continuous.

In this work, the authors propose a novel study into the application of iterative rewriting planning schemes in discrete spaces and evaluate their approach on two tasks: job scheduling and expression simplification. They formulate the rewriting task as a reinforcement learning problem where the action space is the application of a set of possible rewriting rules to modify the discrete state.

The approach is broken down into two steps. In the first step, a particular partition of the discrete state space is selected as needing to be changed by a score predictor. Following this step, a rule selector chooses which action to perform to modify this state space accordingly.

In the job scheduling task, the partition of the state space corresponds to a single job who’s scheduled time must be changed. the application of a rule to rewrite the state involves switching the order of any two jobs to be run. In the expression simplification task, a state to be rewritten corresponds to a subtree in the expression parse tree that can be converted to another expression.

To train, the authors define a mixed loss with two component:
1. A mean squared error term for training the score predictor that minimizes the difference between the benefit of the executed action and the predicted score given to that node
2. An advantage actor critic method for training the rule selector that uses the difference between the benefit of the executed action and the predicted score given to that node as a reward to evaluate the action sampled from the rule set

Pros:

-The approach seems to be relatively novel and the authors address an important problem.
-The authors don’t make their approach more complicated than it needs to be

Cons:

Notation: The notation could be a lot clearer. The variable names used in the tasks should be directly mapped to those defined in the theory in Section 2. It wasn’t clear that the state s_t in the job scheduling problem was defined as the set of all nodes g_j and their edges and that the {\hat g_t} corresponds to a single node. Also, there are some key details that have been relegated to the appendix that should be in the main body of the paper (e.g., how inference was performed)

Evaluation: The authors perform this evaluation on two automatically generated synthetic datasets. It’s not clear that the method would generalize to real data. Why not try the approach on a task such as grammar error correction? Additionally, I would have liked to see more analysis of the method. Apart from showing the comparison of the method with several baselines, the authors don’t provide many insights into how their method works. How data hungry is the method? Seeing as the data is synthetically generated, how effective would the method be with 10X of the training data, or 10% of it? Were any other loss functions attempted for training the model, or did the authors only try the Advantage Actor Critic? What about a self-critical approach? I'd like to see more analysis of how varying different components of the method such as the rule selector and score predictor affect performance.

---

> ### Author Response · Authors · 2018-11-11
> **Response and revision plan**
>
> Thank you for your review and suggestions! We are working on more tasks and ablation study, and will include the results once we finish the experiments. About your questions, we are not sure about what you mean by “self-critical approach”, could you elaborate it?

---

> > ### Comment · AnonReviewer1 · 2018-11-21
> > **self-critical approach**
> >
> > https://arxiv.org/pdf/1612.00563.pdf

---

> > > ### Author Response · Authors · 2018-11-27
> > > **Thank you for your response**
> > >
> > > Thank you for your response! We have tried the self-critical approach, and we find that it does not considerably affect the performance, thus we did not include the results in our revision.

---

### Official Review · AnonReviewer4 · 2018-11-11
**Interesting read but unclear contribution/implications**

**Rating:** 5
**Confidence:** 3

**Review:**

This paper addresses the challenges of prediction-based, progressive planning on discrete state and action spaces. Their proposed method applies existing DAG-LSTM/Tree-LSTM architectures to iteratively refine local sections in the existing plan that could be improved until convergence. These models are then evaluated on a simulated job scheduling dataset and Halide expression simplification.

While this paper presents an interesting approach to the above two problems, its presentation and overall contribution was pretty unclear to me. A few points:

1. Ambiguous model setup: It may have been more advantageous to cut a large portion of Section 3 (Problem Setup), where the authors provide an extensive definition of an optimization problem, in favor of providing more critical details about the model setup. For example, how exactly should we view the job scheduling problem from an RL perspective? How are the state transitions characterized, how is the network actually trained (REINFORCE? something else?), is it episodic (if so, what constitutes an episode?), what is the exploration strategy, etc. It was hard for me to contextualize what exactly was going on

2. Weak experimental section: The authors mention that they compare their Neural Rewriter against DeepRM using a simplified problem setup from the original baseline. I wonder how their method would have fared against a task that was comparable in difficulty to the original method -- this doesn’t feel like a fair comparison. And although their expression simplification results were nice, I would also like to know why the authors chose to evaluate their method on the Halide repository specifically. Since they do not compare their method against any other baselines, it’s hard for me to gauge the significance of their results.

3. Variance across initializations: It would have been nice to see an experiment on how various initializations of schedules/expressions affect the policies learned. I would imagine that poor initializations could lead to poor results, but it would be interesting if the Neural Rewriter was robust to the quality of the initial policy. Since this is not addressed in the paper, it is difficult to gauge whether the authors’ model performed well due to an unfair advantage. Additionally, how much computational overhead is there to providing these (reasonable) initial policies as opposed to learning from scratch?

4. Unclear notation: As previously addressed by other reviewers, key definitions such as the predicted score SP(.) are missing from the text.

---

> ### Author Response · Authors · 2018-11-27
> **Response and clarification**
>
> Thank you for your review! About your questions and comments:
>
> 1. In our job scheduling problem setup:
>
> - Each state is the current job schedule (Figure 2 (a) on page 5).
>
> - The state transition is performed by a rewriting step, which switches the scheduling order of 2 jobs (Section 3.1 on page 3 and Figure 2 (a) on page 5).
>
> - The model is trained with Advantage Actor-Critic algorithm (Section 4.4 on page 6).
>
> - It is episodic, and we consider a rewriting process that starts from an initial schedule for a given set of jobs (earliest job first in our evaluation), and ends at the timestep when the neural network considers that the current schedule cannot be further improved (i.e., the score predictor (SP) computes a negative value), as an episode (Section 5.1.2 on page 7, and more details are in Appendix D on page 14).
>
> - We use epsilon-greedy exploration strategy, and the details are in Appendix D on page 14-15. We can move it to the main body if it is clearer.
>
> 2. In our evaluation:
>
> - The criticism about unfair comparison against DeepRM is incorrect. We evaluate on not only the same tasks as in DeepRM, but also on more complicated settings with larger number of resource types (Section 5.1.1 on page 7). The point is to show our proposed approach is able to deal with more complicated settings than prior works, achieving stronger performance.
>
> - The reasons why we choose to evaluate on Halide repository are two-fold: (1) Halide is widely used at scale in multiple products of Google (e.g., YouTube) and Adobe Photoshop. Its expression simplifier has been carefully tuned with manually-designed heuristics, thus provides a strong baseline for comparison. (2) The format of Halide expressions is general and covers a large part of common operations, including standard arithmetic operators (+, -, *, /, %), boolean operators (&&, ||, !), comparison operators (<, <=, !=), min/max operators, etc (Section 3.2 on page 3, and more details are in Appendix A on page 11). Notice that this is a more comprehensive operator set than previous works on finding equivalent expressions, which consider only boolean expressions [1] [2] or a subset of algorithmic operations [1]. More related work can be found in Section 2 on page 2. Thus, the effectiveness of our approach in the Halide domain provides a good indication that it could also generalize to other expression simplification problems. We have revised Section 3.2 (page 3) to make this point clearer.
>
> - Besides the Halide rewriter, we have added an evaluation on Z3, which is a high-performance theorem prover developed by Microsoft Research. Note that Z3 simplifier works by traversing each sub-formula in the input expression and invoking the solver to find a simpler equivalent one to replace it, thus the simplification steps performed by this solver may not be included in the Halide ruleset, which makes it a strong baseline to compare with. The results and discussion can be found in Section 5.2 (page 8-9).
>
> 3. The concrete rewriting process varies with different initial solutions, e.g., a nearly optimal solution would require a much fewer rewriting steps; however, the quality of the final solution does not heavily depend on the initial one. We have added an ablation study about this point in Appendix E (page 15) in our revision, and we address the main confusion below:
>
> - For job scheduling, we note that the initial schedules are constructed using the earliest-job-first policy, because this schedule is intuitive, easy to compute with a negligible overhead, while is much less effective than the optimal solution, as reported in the paper (Table 1 on page 7). Thus, our evaluation demonstrates that our approach dramatically improves the quality of an initial highly ineffective solution, and results in better ones than computed using other baselines. In our ablation study with initial schedules of different average slow down, the results demonstrate that our neural rewriter model consistently achieves a better performance than baseline approaches. This demonstrates that our rewriting model is robust to the quality of the initial solution.
>
> - For expression simplification, since our evaluation metric is the average reduction of expressions (Section 5.2.1 on page 8), the results demonstrate that our approach significantly reduces the complexity of the initial expressions (Table 3 on page 8). Note that the initial expressions could be quite complicated, e.g., with a parse tree of 100 nodes (Table 2 on page 8).
>
> 4. These definitions are in Section 4.1 (page 4).
>
> [1] Miltiadis Allamanis, Pankajan Chanthirasegaran, Pushmeet Kohli, Charles Sutton, Learning Continuous Semantic Representations of Symbolic Expressions, ICML 2017.
> [2] Richard Evans, David Saxton, David Amos, Pushmeet Kohli, Edward Grefenstette, Can Neural Networks Understand Logical Entailment? ICLR 2018.

---

### Author Response · Authors · 2018-11-27
**Revision**

We thank all reviewers for their comments! We have revised the paper with the following major changes to incorporate the comments:

- We have added an ablation study to demonstrate that our approach is not heavily biased by the initial solutions.

- For expression simplification, we have added an evaluation on Z3, a high-performance theorem prover developed by Microsoft Research. Since its simplifier would invoke a solver to rewrite the expressions, the simplification steps performed by this solver may not be included in the Halide ruleset, which makes it a strong baseline to compare with.

---

### Meta-Review · Area_Chair1 · 2018-12-13
**Borderline paper**

**Confidence:** 3
**Recommendation:** Reject

**Metareview:**

This paper provides a new approach for progressive planning on discrete state and action spaces. The authors use LSTM architectures to iteratively select and improve local segments of an existing plan. They formulate the rewriting task as a reinforcement learning problem where the action space is the application of a set of possible rewriting rules. These models are then evaluated on a simulated job scheduling dataset and Halide expression simplification. This is an interesting paper dealing with an important problem. The proposed solution based on combining several existing pieces is novel. On the negative side, the reviewers thought the writing could be improved, and the main ideas are not explained clearly. Furthermore, the experimental evaluation is weak.